# An Evaluation of Continuing Medical Education among Pharmacists in Various Pharmacy Sectors in the Asir Region of Saudi Arabia

**DOI:** 10.3390/healthcare11142060

**Published:** 2023-07-19

**Authors:** Geetha Kandasamy, Dalia Almaghaslah, Mona Almanasef

**Affiliations:** Department of Clinical Pharmacy, College of Pharmacy, King Khalid University, Abha 61421, Saudi Arabia; damoazle@kku.edu.sa (D.A.); malmanasaef@kku.edu.sa (M.A.)

**Keywords:** continuing medical education, continuing professional development, community pharmacy, Saudi Arabia, pharmacist

## Abstract

Background: Continuing medical training is an important component of modern medical practice because it maintains the ability of physicians to provide up-to-date patient care. This study explored pharmacists’ involvement in CME activities and investigated the barriers to undertaking CME activities in Saudi Arabia. It also aimed to highlight the obstacles that prevent pharmacists from participating in CE activities. Methods: This study used a cross-sectional self-administered web-based questionnaire. It was conducted among practising pharmacists in various pharmacy fields in the Asir region of Saudi Arabia. The structured questionnaire consisted of four domains. A convenience sampling strategy was used to select and recruit study participants. The results were described in terms of frequencies and percentages. A Chi-square test was used to assess the differences for categorical data. *p* value of <0.05 was considered significant. Results and conclusions: A total of 173 pharmacists participated in the study. Attending the conference was the most popular activity (67%), followed by training courses (61.8%) and approved web-based activities (60.1%). Regarding barriers that prevented pharmacists from participating in CME activities, a lack of a scientific database or books was the top-rated barrier that hindered pharmacists from obtaining the required CME hours (79.8%). Another important barrier was the cost of the activities (74.6%). Regarding the area of practice for which pharmacists would be interested in attending CME activities, public health was the favourite (89.6%), followed by personal skills (82.7%). Gathering the required CME hours for re-registration was the main motive for attending CME activities in most of the pharmacy sectors. Policymakers should consider shifting the current CME system to the Continuing Professional Development (CPD) model, which promotes engagement in professional development activities that are relevant to the scope of practice.

## 1. Introduction

The recent decades have seen a significant transformation in the pharmacy profession. The responsibilities of pharmacists now include pharmaceutical care and public health services, in addition to dispensing pharmaceuticals [1]. Due to their experience in identifying, resolving, and preventing medication mistakes and other pharmaceutical-related issues, pharmacists have a large chance to significantly contribute to lowering healthcare costs. There are now more pharmacists working in clinically advanced jobs worldwide due to the recent growth of clinical pharmacy practice. The majority of published research suggests that pharmacist activities are cost effective. Pharmacist-provided services and clinical interventions have been shown to reduce the risk of potential adverse drug events and improve patient outcomes [2]. According to the consensus reached at the College of General Practice of Canada’s first conference on medical education held in Toronto in the fall of 1962, Continuing Medical Education (CME) is an essential feature of the practice of modern medicine, as it maintains the doctor’s ability to provide quality patient care [3].

Continuing medical training is an important component of modern medical practice because it maintains the ability of physicians to provide up-to-date patient care. The undergraduate medical student must actively cultivate the idea of “lifelong learning”, and the graduate doctor must maintain it, as both a practitioner and a teacher [4]. According to the Accreditation Council for Pharmacy Education, CE (Continuous Education) is an organised educational activity that supports pharmacists’ ongoing professional growth in order to preserve and improve their competence [5]. Previous research has demonstrated that drug information resources are insufficient [6]. Hence, structured CME programmes are crucial to advancing the knowledge and mindsets of pharmacists employed in the Kingdom in order for them to fulfil their responsibilities to their patients and the community [7]. Concerns about patient safety increase pharmacist interest in CE, and recent research has shown a correlation between participation in CE and relative self-motivation [8]. Healthcare workers can keep abreast of the latest scientific advancements, improve their clinical expertise, and maintain awareness of the newest social concerns through effective lifelong learning systems [9,10].

CPD is defined by the International Pharmaceutical Federation (FIP) as “the responsibility of individual pharmacists for systematic maintenance, growth, and widening of knowledge, skills, and attitudes, to ensure continued competency as a professional, throughout their career” [11]. Due to technological innovation, significant changes in the roles and responsibilities of pharmacists have been noted. One dynamic of the profession is a global transition from patient-oriented to product-oriented services. Pharmacy practitioners must maintain competence throughout their careers and refresh their knowledge and ability to carry out additional tasks successfully in order to face these new difficulties. This demonstrates the necessity of Continuing Professional Development (CPD) programmes in assisting pharmacy practitioners’ roles across a range of healthcare settings [12]. Governments and insurance companies worldwide have suffered as a result of drastically growing pharmaceutical costs. Therefore, promoting medications by their generic names and encouraging pharmacists to fill prescriptions with generic medications reduces the use of healthcare services and associated costs [13,14]. Continuing education programs (CEPs) with the aid of educational training/practicum have been suggested as a vital tool for health care professionals, including pharmacists, to enhance patient outcomes, with the best results in clinical outcomes demonstrated [15]. Previous studies have shown that CEPs have beneficial effects on the performance, attitudes, and knowledge of healthcare professionals [16]. Previous research on generic drug knowledge of pharmacy students indicated knowledge gaps [17].

Studies conducted in Saudi Arabia and other nations revealed that pharmacists and medical professionals lack sufficient knowledge and/or trust. Each nation has created its own method of achieving CE and CPD. For the renewal of a pharmacist’s license, many nations, including France and the United States, now require CE with a set number of credits (US). Mandatory CPD is also practiced in the UK and Canada. Jordan and Iran in the Middle East both have CE systems that are required. Since 1991, continuing education has been mandated in Iran, with pharmacists needing to complete 25 h of CE annually in order to renew their licenses. Saudi Arabia and the United Arab Emirates (UAE) are the two nations in the Arabian Gulf where CE systems are required [9]. The SCFHS governs the licencing and re-registration of pharmacists in Saudi Arabia [18]. SCFHS uses Mumaris Plus (Mumaris+), an electronic gateway, to register and renew pharmacist licences to practice pharmacy. The site displays the CME credits given to pharmacists who participate in educational events presented by organisations that have been granted SCFHS accreditation. Every two years, pharmacists are required to complete 40 CME hours. There are two categories from which to choose these CME hours: the first category offers up to 25 h for attending conferences, seminars, workshops, training sessions, research, journal articles, and book publications. The second type consists of general workshops, authorised web-based activities, and internal activities that can last up to 15 h [19]. CPD for pharmacists is essential to educate them about and promote the broad use of generic medications, particularly for future healthcare providers, as their knowledge of generic medications can affect their future with respect to prescriptions and trends in generic substitution [20].

A pilot study demonstrated the advantages of creating CPD packages for pharmacy professionals in the future, which would focus on improvements in pharmacy practices and skill development, use in-person presentations and handouts, and concentrate on the participant’s curiosity to advance their professional practice [21]). For a pharmacist to keep their knowledge and abilities up to date, CE is essential throughout their career [22]. A randomised controlled trial demonstrated the effectiveness of several types of continuing education meetings for community pharmacists who offer weight control treatments [23].

This study explores pharmacists’ involvement in CME activities and investigated the barriers to undertaking CME activities in Saudi Arabia. It also aims to highlight the obstacles that prevent pharmacists from participating in CE activities.

## 2. Methodology

### 2.1. Study Design

The study was conducted between November 2021 and November 2022. This study followed a cross-sectional design using an anonymous self-administered web-based questionnaire. It was conducted among practising pharmacists in various pharmacy fields in the Asir region of Saudi Arabia.

### 2.2. Population and Settings

This study was conducted with pharmacists practicing in different pharmacy sectors in Saudi Arabia, including community pharmacy, hospital pharmacy, regulatory, academia and the pharmaceutical industry. The total number of registered pharmacists in Saudi Arabia reached 29,900 in 2018. The inclusion criteria: Qualified pharmacists working in different pharmacy sectors in the Asir region, Saudi Arabia. Exclusion criteria: Pharmacy technicians, unemployed pharmacists, pharmacists working outside the Asir region.

### 2.3. Sample Size and Sampling Procedure

The sample size was based on the total number of pharmacists in the Asir region of Saudi Arabia n = 747 and determined by using the Raosoft online sample size calculator with a predetermined margin of error of 5% and a confidence level of error of 95%. In order to minimise errors in the findings and increase study reliability, the target sample size was set at n = 254. Of a sample of n = 254, only 173 agreed to participate in the study. A convenience sampling strategy was used to select and recruit study participants. 

### 2.4. Data Collection Tool

The questionnaire was adapted from previous studies [18,19]. The structured questionnaire consisted of four domains. Section 1 collected demographics and background information including age, gender, nationality, area of practice, qualification, work experience, and registration at the SCFHS. Section 2 gathered data related to the type of continuing medical education activities attended and asked whether participants received CME points, chose those activities relevant to their practice, or based their choice on personal development needs or to meet the CME hours required for the SCFHS re-registration. Conference attendance, seminar attendance, workshops, training courses, book writing, scientific paper publication, conducting research, reviewing scientific research, internal activities, panel discussion, general workshops, and web-based approved activities were included. Section 3 aimed to identify the barriers that prevented pharmacists from participating in CME activities. Issues such as lack of scientific databases/books, irregular conference organisation, lack of personal time and motivation, difficulty to take a leave from work to attend the CE activity/lack of staff, lack of announcements and advertising for CE activities in Saudi Arabia, cost of CE activities, lack of distance learning methods, the fact that the conducted CE workshops are outdated and not effective, lack of a national competency framework for pharmacist knowledge to identify personal improvement areas, the belief that one’s practice does not need to attend CPD activities were all included. Section 4 determined pharmacists’ professional development needs. Professional practice, public health, supply of medicine, safe and rational use of medicine, organization and management skills, and personal skills were all included factors.

The face and content validation was carried out by two academics from the College of Pharmacy, King Khalid University, who have expertise in the field of continuing medical education.

The questionnaire was available in English. The final questionnaire was piloted with five pharmacists practising in different pharmacy settings to ensure the clarity of the language and questionnaire structure. The results of the pilot were not included in the final results.

The questionnaire was constructed using Google Forms and was distributed online through multiple channels, i.e., official institutional emails and through social media platforms. Monthly reminders were sent to the potential participants with an attempt to maximize response rate. 

### 2.5. Statistical Analysis

Data collected was downloaded, coded, entered and analysed using SPSS version 25 for Mac. The results are described in terms of frequencies and percentages. A Chi-square test was used to assess differences for categorical data. A *p* value of < 0.05 was considered significant.

### 2.6. Ethics Approval

The ethical approval was given by the Ethical Committee of Scientific Research, King Khalid University EMC#2021-5504. All respondents were asked for their written consent prior to participation in the study. Participation in the study was voluntary, and participants had the right to decline the invitation to participate without any penalty.

## 3. Results

Table 1 shows the demographics and background information. A total of 173 pharmacists participated in the study. Most pharmacists (78.1%) were between the ages of 21 and 39. Around two thirds (61.8%) were female and 76.3% were Saudi nationals. Under half of participants (42.2%) were practising in institutional settings, and under one third were academics; 20.8% were working in community pharmacy. A minority worked in the regulatory industry (5.2%), and 4.6% worked in the pharmaceutical industry. More than half (55.5%) of the pharmacists held a first degree in pharmacy, just under a quarter (24.3%) held a Master’s degree in pharmacy, and 20% held a PhD in pharmacy. Approximately half of the pharmacists were fresh graduates who have been working in the field of pharmacy for less than 5 years, less than a third (31.8%) have been working for 5 to 10 years, and (22%) have been working for more than 10 years. More than three quarters of the participants (75.1%) were SCFHS.

Table 2 shows the type of CME activities the participants attended. Attending conference was the most popular activity (67%), followed by training courses (61.8%), approved online activities (60.1%) and seminars and internal activities scored similar scores with 54.9% and 53.8%, respectively. Conducting research, publishing articles and reviewing scientific research scored 50.9%, 45.1% and 44.5%, respectively. Panel discussions and workshops obtained 48% and 41.6%, respectively. The least popular activities were general workshops and book writing with 33.5% and 25.4%, respectively.

Table 3 shows the barriers that prevented pharmacists from taking part in CME activities. A lack of scientific databases or books was the top barrier preventing pharmacists from obtaining the required CME hours (79.8%). Other important barriers were the cost of activities (74.6%), difficulty of taking leave from work due to staff shortages (74%), lack of online activities (72.8%), lack of announcements and advertising of CE activities in Saudi Arabia (72.8%), practice stings that do not require pharmacists to be licensed (71.7%), the lack of a national competency framework concerning pharmacist knowledge to identify personal improvement areas (69%). Less important barriers were the nonavailability of regular conferences (61.8%), lack of personal time (58.4%), and the perception that CE workshops are outdated and ineffective (58.4%).

Table 4 shows the area of practice where pharmacists would be interested in participating in CME activities. Public health was the preferred area of practice with 89.6%, followed by personal skills with 82.7%, organisation and management skills with 82.1%, supply with 79.2% and safe and rational use of medications with 75.7%. The least preferred area was professional practice, with 42.2%.

Table 5 shows the information relating to CME hours by pharmacy sector. For the majority of pharmacists practising in community pharmacies, the main motive for attending CME activities was obtaining the required hours for re-registration (83.33%). Three quarters (72.2%) indicated that the activities they attended were based on personal needs, and around the same number reported that they were awarded the hours for the activities attended. Just over three quarters (77.8%) suggested that CME activities were relevant to their practice. A total of 78% of hospital pharmacists attended CME activities to maintain their registration at SCFHS and around the same number reported that the activities were relevant to their practice and were based on personal development needs, while 72.6% were awarded the hours for the activities. Achieving the required hours was a less important motive for academics, with 59.6% finding that relevance to practice was a more important factor. Around 72% of academic staff attended activities that were based on personal development needs and 74.5% were awarded CME hours for the activities they participated in. Three quarters of pharmacists working in the pharmaceutical industry found that achieving the required hours was equally important to participating in activities that were relevant to their practice area. Personal development needs were found to be less important in this sector, with (62.5%), and fewer pharmacists were awarded hours for the activities they attended.

Table 6 shows pharmacist registration at the SCFHS by pharmacy sector and nationality. The majority of pharmacists in all sectors were licensed to practice pharmacy in Saudi Arabia. The highest percentage of registered pharmacists was found in the regulatory sector with 88.9%, the pharmaceutical industry with 87.5%, hospital pharmacies (80.8%) and community pharmacies (80.5%). Academia recorded the lowest percentage of registered pharmacists with 57.4% (*p* = 0.024). The number of registered Saudi pharmacists (78.8%) was significantly higher than that of non-Saudi nationals (63.4%) (*p* = 0.04).

## 4. Discussion

This study explored pharmacist involvement in CME activities and investigated the barriers to undertaking CME activities in Saudi Arabia. Ensuring a competent pharmacy workforce is essential to provide safe and effective health services and outcomes [19,24]. Biennial completion of 40 CME hours is required by SCFHS for pharmacist registration and re-licensing [19,25]. Current research found that conferences were the most popular activity attended by study participants, followed by attending training courses and approved Web-based activities. The popularity of conferences could be due to the fact that they are usually run on multiple days with diverse programmes and therefore offer a high number of CME hours and knowledge enrichment. In addition, conferences provide a chance to network with professionals in the same or similar fields and the exchange of knowledge and experience. The largest annual pharmacy conference in the country, SIPHA (Saudi International Pharmaceutical Sciences Meeting and workshop), is organised by the Saudi Pharmaceutical Society and awards 30 CME hours, which is equivalent to 2/3 of the total CME hours required for registration. This could also be another reason for participants’ preference. Participating in online CME activities is probably favoured for being convenient and cost effective [26]. When arranged in ascending order, the most preferred area for CME was public health, followed by personal skills, organisation and management skills, medication supply, safe and rational use of medicines and, finally, professional practice. Similar findings were reported in a recent study confirming that pharmacists practicing in Saudi Arabia found that pharmaceutical public health and personal/professional behaviours are the most relevant to their practice. This is explained by the fact that most of the participants have been practising for more than 5 years and therefore have focused on these areas of practice as they move up the career ladder [27].

Obtaining the CME hours required for re-registration was the main motive for attending CME activities among pharmacists in all the pharmacy sectors, with the exception of academia, which is derived by activity relevance to practice. The previous literature suggested that academics, especially those in the department of pharmaceutical sciences, were less interested in pharmaceutical care and public health as they are solely involved in teaching and research-based activity [27]. In Saudi Arabia, a professional licence is not currently required for academics who are not involved in pharmacy practice work, unlike any other pharmacy sector that requires a professional licence, such as hospital or community pharmacies. However, it is worth noting that this licence is required from academics for participating as speakers in national conferences.

Our findings are in agreement with those of previous work, which suggests that the scope of practice of the pharmacist is not really relevant to their registration and re-licensing requirements, nor are these linked to the identified learning needs of the individual [4,19,28,29,30]. Additionally, reflection is not required in the current Saudi CME system, and attending unplanned activities does not fit with the registration requirements, and is therefore not normally accepted [19]. Gathering the requires credit hours for re-registration would probably be the main incentive for attending CME activities, which could defeat the purpose of professional development and result in negative consequences on the quality healthcare services offered by the pharmacist. Thus, the current CME system would encourage pharmacists to participate in non-preferred activities in order to achieve the credit hours required for re-registration. Previously published research highlights that the continuing professional development of the pharmaceutical workforce in Saudi Aribia is substandard [25]. Another recent study calls for the current CME system to be shifted to a CPD model that allows pharmacists to customise their CPD plans according to their needs and scope of practice [19]. The current study did not aim to offer guidelines on how to convert the current CME system into a CPD model and investigate the enables and barriers for the shift; however, this could be a priority for future work.

Among the top-rated barriers that prevented pharmacists from participating in CME activities were the lack of scientific databases or books, difficulty in taking leave from work due to staff shortages, lack of online activities, lack of announcements and advertising of CE activities, practice settings not requiring pharmacists to be licenced, and lack of a national competency framework for pharmacist knowledge to identify personal improvement. In Kuwait, similar barriers were highlighted with the lack of personal time being the top barrier followed by the lack of scientific books and databases and limited national conferences [31]. An earlier study suggests the first competency framework for Saudi Arabia’s entry-level pharmacists. In order to meet the needs of the nation for pharmaceutical services, the developed framework represents a consensus on competencies for foundation-level pharmacists working across all pharmacy sectors and is eligible for supporting further improvement of initial pharmacy education. It also supports excellence in pharmacist performance [32].

Shifting the current CME system to a CPD model that takes into consideration reflexion and attending unplanned activities will probably tackle some of the barriers identified above. Having a CME model that is not linked to a need-based health initiative was previously highlighted as an issue of workforce development need in Saudi Arabia [25]. Additionally, Saudi Arabia should establish a national framework that aims to assist pharmacists to distinguish their areas of personal improvement areas. Therefore, participation in activities would be mapped with relevant competencies and based on personal needs. Future research could include investigating current issues using qualitative research methods such as focus groups or interviews to better understand the experiences and views of pharmacists. Additionally, due to the nature of the study design, the generalizability of the findings is limited to those with a similar context.

## 5. Limitations

The current study has some limitations which should be considered when interpreting the study findings. First, the number of participants from some sectors—i.e., the regulatory and pharmaceutical companies—is relatively small. In addition, the majority (78%) of participants have less than ten years of experience. This could be attributed to the fact that data was collected using a convenience sampling strategy which could have also introduced bias and reduced the generalizability of the study findings. However, this study provides a snapshot of the issues with the CME system in Saudi Arabia amongst a sample of pharmacists practicing in different sectors. Although the minimum recommended sample size was not achieved, we do not presume that this might have affected the overall conclusions of this research.

## 6. Conclusions

The current study illustrated the common issues associated with the present CME system among pharmacists practising in different sectors in Saudi Arabia. Gathering the required CME hours for re-registration was the main motive for attending CME activities in most of the pharmacy sectors. Policymakers should consider shifting the current CME system to the CPD model, which promotes engagement in professional development activities that are relevant to the scope of practice of the pharmacist and linked to the identified learning needs.

## Figures and Tables

**Table 1 healthcare-11-02060-t001:** Demographics and background information.

	n	%
Age		
21–29	65	37.6
30–39	70	40.5
40–49	24	13.9
50+	14	8.1
Gender		
Male	66	38.2
Female	107	61.8
Nationality		
Saudi	132	76.3
Non-Saudi	41	23.7
Pharmacy Sector		
Community pharmacy	36	20.8
Hospital pharmacy	73	42.2
Academia	47	27.2
Regulatory	9	5.2
Pharmaceutical companies	8	4.6
Qualification		
First degree in pharmacy (Bachelor of Pharmacy, Pharm D)	96	55.5
Master of Pharmacy	42	24.3
Doctor of Pharmacy (PhD)	35	20
Experience		
<5 years	80	46.2
5 to 10 years	55	31.8
More than 10 years	38	22
SCFHS registration		
Registered	130	75.1
Not registered	43	24.9

SCFHS—Saudi Commission for Healthcare Specialities.

**Table 2 healthcare-11-02060-t002:** CME activities attended.

Type of Activities	n	%
Attendance of conferences	117	67.6
Seminar attendance	95	54.9
Workshops	72	41.6
Training courses	107	61.8
Books writing	44	25.4
Publication of scientific papers	78	45.1
Conducting research	88	50.9
Reviewing scientific research	77	44.5
Internal activities	93	53.8
Panel discussions	83	48
General workshops	58	33.5
Approved Web-based activities	104	60.1

CME—Continuing Medical Education.

**Table 3 healthcare-11-02060-t003:** The barriers that prevented pharmacists from taking part in CME activities.

Barriers	n	%
Unavailability of scientific databases/books	138	79.8
Conferences are not regularly organised	107	61.8
Lack of personal time and motivation	101	58.4
Difficulty of taking leave from work to attend the CE activity/lack of staff	128	74
Lack of announcement and advertising of CE activity in Saudi Arabia	126	72.8
Cost of CE activity	129	74.6
Lack of distance learning methods	126	72.8
CE workshops are outdated and ineffective	101	58.4
Lack of a national competency framework concerning pharmacists’ knowledge to identify personal improvement areas	109	69
My practice does not require attendance of CPD activities	124	71.7

CME—Continuing Medical Education, CE—Continuous Education.

**Table 4 healthcare-11-02060-t004:** Area of practice where pharmacists would be interested in attending CME activities.

Area of	n	%
Professional Practice	73	42.2
Public health	155	89.6
Supply of Medicines	137	79.2
Safe and rational use of medicines	131	75.7
Organisation and management skills	142	82.1
Personal skills	143	82.7

CME—Continuous Medical Education.

**Table 5 healthcare-11-02060-t005:** CME activities by pharmacy sector.

CME Activities by Pharmacy Sector	Relevant to Practice	Based on Personal Needs	CME Rewarded	Achieving Hours
Community Pharmacy	77.78	72.22	72.22	83.33
Hospital Pharmacy	78.08	79.45	72.6	78.08
Academia	82.98	72.34	74.47	59.57
Regulatory	66.67	77.78	77.78	77.78
Industry	75	62.5	62.5	75

CME—Continuing Medical Medication.

**Table 6 healthcare-11-02060-t006:** Pharmacy registration at SCFHS by sector and nationality.

	n	%	Chi Square	df	*p* Value
Pharmacy sector					
Community pharmacy	29	80.5	11.3	4	0.024
Hospital pharmacy	59	80.8			
Academia	27	57.4			
Regulatory	8	88.9			
Pharmaceutical companies	7	87.5			
Nationality					
Saudi Arabia	104	78.8	3.9	1	0.04
Non-Saudi	26	63.4			

SCFHS—Saudi Commission for Healthcare Specialities.

## Data Availability

The datasets used and analysed during the current study are available from the corresponding author on reasonable request.

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
