# Peer review of "An Evaluation of Continuing Medical Education among Pharmacists in Various Pharmacy Sectors in the Asir Region of Saudi Arabia"

_healthcare, 2023, doi:10.3390/healthcare11142060_

Round 1

Reviewer 1 Report

Dear author,

I have completed the reviewing process. You touched upon an important issue. However, there are many such studies in the literature. Moreover, since very little data was disclosed as a result, the contribution and originality of the study should be explained by emphasizing. Therefore, there are some grammatical errors. Thus, the article should be checked by a native speaker.

The manuscript requires major revision. It can be re-evaluated after the authors make the necessary adjustments. Recommendations and requirements are described below; 

1. The total number of questions asked to the participants should be explained and the results should be detailed.

2. Please explain the acronym “SCFHS” where it was first used (line 19). Then use only the acronym.

3. Please explain the acronym “CME” where it was first used. Then use only the acronym.

4. In lines 22 and 23, it was written “… most popular activity with (61.8%) and…”, “…activity with (60.1%)..”Please choose parenthesis or “with”. Use “… most popular activity with 61.8% and…” or “… most popular activity (61.8%) and…”

5. In lines 22, 23, and 24 don’t use “the second most popular”, “the third most popular” etc. You can use the phrase of “respectively” or “following by”. When the same words are used over and over, it becomes difficult to understand.

6. Please explain the acronym “CPD where it was first used. Then use only the acronym.

7. Please don’t use the acronyms as keywords.

8. One tab space was used at the beginning of the paragraph in some sentences, while no space was applied in the others. The entire article should be rearranged according to the guidance of the journal.

9. The sample size should be indicated as “n=…” in the sentences. For example; line 134 (n=747), line 137 (n=254).

10. It was stated that the minimum number of subjects to participate in the study was 254 according to the results of Raosoft online sample size calculation. But the study was limited to 173 participants. Why was the study conducted with fewer participants? It should be checked whether working with subjects below the minimum number changes the limits of statistical significance.

11. Statistical analysis should be performed for the data explained in Table 1 and Table 2. P-values should be calculated for each category. A new column should be added to the tables and p-values should be written there.

12. The explanation of the acronym used in each table should be explained below the table.

13. In Figure 1, it is not clear what the x and y axis represent. Relevant explanations should be added to the graph.

14. In line 236, the p-value should be written in parentheses. The correct explanation should be like “…..higher than non-Saudi nationals (63.4%) (p=0.04)”.

15. It cannot be understood what the value written "chi-square" in Table 5 is. Instead of the name of the test, the name of the coefficient should be written.

16. Only pharmacy sector and nationality data were statistically compared. Statistical analysis should be conducted using more data and results should be detailed.

Some sentences have word repetitions.

E.g. Attending conferences was the most popular activity (67%), training courses were the second 22 most popular activity with (61.8%) and Web-based approved activities was the third most popular 23 activity with (60.1%).

There are also some errors in tenses. The manuscript should be reviewed by a native speaker.

Author Response

Thank you for your valuable comments. We have revised our article thoroughly as per your comments. The individual response to reviewer comments are given below.

Manuscript ID: Healthcare-2486693

Reviewer Report 1

Dear author,

I have completed the reviewing process. You touched upon an important issue. However, there are many such studies in the literature. Moreover, since very little data was disclosed as a result, the contribution and originality of the study should be explained by emphasizing. Therefore, there are some grammatical errors. Thus, the article should be checked by a native speaker.

The manuscript requires major revision. It can be re-evaluated after the authors make the necessary adjustments. Recommendations and requirements are described below; 

  1. The total number of questions asked to the participants should be explained and the results should be detailed.

 Response to reviewers: Changes were made as suggested by the reviewer.

The questionnaire was adapted from previous studies [18,19]. The structured questionnaire consisted of four domains. Section One collected demographics and background information including age, gender, nationality, area of practice, qualification, work experience, and registration at the SCFHS. Section Two gathered data related to the type of continuing medical education activities attended, and asked whether participants received CME points, chose those activities relevant to their practice, or based their choice on personal development needs or to meet the CME hours required for the SCFHS re-registration. Conferences attendance, seminar attendance, workshop, training courses, book writing, scientific papers publication, conducting research, reviewing scientific research, internal activities, panel discussion, general workshops, web-based approved activities. Section 3 aimed to identify the barriers that prevented pharmacists from participating in CME activities. Questions such as; Lack of Scientific databases/books are not available. Conferences are not regularly organized. Lack of personal time and motivation. Difficulty to take a leave from work to attend the CE activity/ lack of staff. Lack of announcements and advertising for CE activity in Saudi. Cost of the CE activity. Lack of distance learning methods. The conducted CE workshops are outdated, not effective.  Lack of a national competency framework for pharmacists’ knowledge to identify personal improvement areas. My practice does not need to attend CPD activities. Section Four determined pharmacists’ professional development needs. Professional practice, public health, supply of medicines, safe and rational use of medicines, organization and management skills, and personal skills.

Results were detailed in all the tables.

  1. Please explain the acronym “SCFHS” where it was first used (line 19). Then use only the acronym.

Response to reviewers: Acronym Explained in line 19. Saudi Commission for Healthcare Specialities.

  1. Please explain the acronym “CME” where it was first used. Then use only the acronym.

Response to reviewers: Acronym explained. ie Continuing Medical Education

  1. In lines 22 and 23, it was written “… most popular activity with (61.8%) and…”“…activity with (60.1%).”.  Please choose parenthesis or “with”. Use “… most popular activity with 61.8% and…” or“… most popular activity (61.8%) and…”

Response to reviewers: Changes done as suggested by the reviewer according to the comments on the quality of English language (repetition of words ie most popular activity)

  1. In lines 22, 23, and 24 don’t use “the second most popular”, “the third most popular” etc. You can use the phrase of “respectively” or “following by”. When the same words are used over and over, it becomes difficult to understand.

Response to reviewers: Changes done as suggested by the reviewer. Attending conference was the most popular activity (67%), followed by training courses (61.8%) and web based approved activities (60.1%).  

  1. Please explain the acronym “CPD”where it was first used. Then use only the acronym.

Response to reviewers: Acronym CPD explained ie Continuing Professional Development

  1. Please don’t use the acronyms as keywords.

Response to reviewers: Acronym has been removed

  1. One tab space was used at the beginning of the paragraph in some sentences, while no space was applied in the others. The entire article should be rearranged according to the guidance of the journal.

Response to reviewers: One tab Space given throughout the manuscript as per the guidance of the journal.

  1. The sample size should be indicated as “n=…” in the sentences. For example; line 134 (n=747), line 137 (n=254).

Response to reviewers: Changes have been done according to reviewers suggestion.

  1. It was stated that the minimum number of subjects to participate in the study was 254according to the results of Raosoft online sample size calculation. But the study was limited to 173 participants. Why was the study conducted with fewer participants? It should be checked whether working with subjects below the minimum number changes the limits of statistical significance.

Response to reviewers: Thank you very much for pointing this out. To clarify, the minimum recommended sample size was 254 participants, the sampling strategy used in this research was convenience sampling. Participants were invited to participate in the study through multiple channels i.e., official institutional emails and social media platforms. Monthly reminders were sent to the potential participants with an attempt to maximize the response rate. However, only 173 participants agreed to take part in the study and completed the questionnaire. Modifications were made to the methods and discussion section to address this comment. 

Changes made in the manuscript:

Sample size and sampling procedures

A convenience sampling strategy was used to select and recruit the study participants. 

Data collection form 

Monthly reminders were sent to the potential participants with an attempt to maximize the response rate. 

Limitations

Although the minimum recommended sample size was not achieved, we do not presume that this might have affected the overall conclusions of this research. 

  1. Statistical analysis should be performed for the data explained in Table 1 and Table 2. P-values should be calculated for each category. A new column should be added to the tables and p-values should be written there.

Response to reviewers: Descriptive statistics was chosen to analyse demographic data. It is unusual to run a p value for demographics 

  1. The explanation of the acronym used in each table should be explained below the table.

Response to reviewers: Explanation of the acronym given below the table.

  1. In Figure 1, it is not clear what the x and y axis represent. Relevant explanations should be added to the graph.

Response to reviewers: Thank you for points this out. X refers to pharmacists by practice sector. Y refers to CE activities attended. According to reviewer (2) comments figure 1 has been presented in Table. 

  1. In line 236, the p-value should be written in parentheses.The correct explanation should be like “…..higher than non-Saudi nationals (63.4%) (p=0.04)”.

Response to reviewers:  Changes were done as suggested by the reviewer

The number of registered Saudi pharmacists (78.8%) was significantly higher than non-Saudi nationals (63.4%) (p =0.04).

  1. It cannot be understood what the value written "chi-square" in Table 5 is. Instead of the name of the test, the name of the coefficient should be written.

Response to reviewers: This how chi square is presented (chi square) refers to the test statistics. 
16. Only pharmacy sector and nationality data were statistically compared. Statistical analysis should be conducted using more data and results should be detailed.

Response to reviewers: The SCFHCS doesn't consider parameters such as age and gender for pharmacists re-licensing for that reason, we only consider the variables that would have a meaningful comparison and would affect pharmacy practice nationally. 

Comments on the Quality of English Language : Some sentences have word repetitions.

E.g. Attending conferences was the most popular activity (67%), training courses were the second 22 most popular activity with (61.8%) and Web-based approved activities was the third most popular 23 activity with (60.1%).

Response to reviewers: Changes were made as suggested by the reviewer. Attending conference was the most popular activity (67%), followed by training courses (61.8%) and web based approved activities (60.1%).

There are also some errors in tenses. The manuscript should be reviewed by a native speaker.

Response: We have addressed all typos and English language extensively and thoroughly throughout Grammarly and Writefull in the manuscript as per the Reviewer’s Comments.

Once again, thank you to all the reviewers for the valuable comments and suggestions. We have revised our manuscript strictly and thoroughly as per your comments for more clarity to the reader.

Reviewer 2 Report

Thank you for giving me a chance to review this study. It is a good descriptive study that present a general picture of CME among pharmacists in Asir, Saudi Arabia.

1. In the abstract and after the Background please write the aim of study.

2. Instead of writing the details of questionnaire in Abstract please write about sampling and statistical test used for analysing.

3. Please use the full words of CME and SCFHS and then use their abbreviation in Abstract.

4. In the last paragraph of Introduction, please revise the purpose of study. It is not to understand pharmacist attitude towards CE.

5. Please move and combine the study period (section 2.4) to the section 2.1.

6. In the section 2.2, please clearly mention what are the inclusion and exclusion criteria? Please also justify why this study is conducted in Asir? Any particular reason?

7. In the section 2.3, please write what was the sampling method? Was it random probability sampling or non-probability sampling? Which type?

8. In the section 2.5, please write “data collection tool” instead of ‘data collection form”.

9. In the section 2.5, please write how did you develop the questionnaire? Was it already developed? Or you generate it suing other similar questionnaire?

10. Please write about study procedure. How did you reach the study participants? How long it took time the questionnaire to be completed? How about the period (time interval) to complete the questionnaire? How about non-respondent rate?

11. Please present the results of Figure 1 in a table.

12. Please write about lack of generalizability of finding to all Saudi Arabia, as one of the limitations of the study due to nature of study design.

None

Author Response

Thank you for your valuable comments. We have revised our article thoroughly as per your comments. The individual response to reviewer comments are given below.

Manuscript ID: Healthcare-2486693

Review report 2

Comments and Suggestions for Authors

Thank you for giving me a chance to review this study. It is a good descriptive study that present a general picture of CME among pharmacists in Asir, Saudi Arabia.

1.In the abstract and after the Background please write the aim of study.

Response to reviewers: Changes has been incorporated as suggested by the reviewer. This study has explored pharmacists’ involvement in CME activities and investigated the barriers to undertaking CME activities in Saudi Arabia. It also aims to highlight the obstacles that prevent pharmacists from participating in CE activities.

  1. Instead of writing the details of questionnaire in Abstract please write about sampling and statistical test used for analysing.

Response to reviewers: A convenience sampling strategy was used to select and recruit the study participants.  The results are described in terms of frequencies and percentages. A Chi square test was used to assess differences for categorical data. P value <0.05 was considered significant.

  1. Please use the full words of CME and SCFHS and then use their abbreviation in Abstract.

Response to reviewers: The full words of CME and SCFHS are explained in the abstract.

  1. In the last paragraph of Introduction, please revise the purpose of study. It is not to understand pharmacist attitude towards CE.

Response to reviewers: This study has explored pharmacists’ involvement in CME activities and investigated the barriers to undertaking CME activities in Saudi Arabia.

  1. Please move and combine the study period (section 2.4) to the section 2.1.

Response to reviewers: Section 2.4 combined with Section 2.1 as suggested by the reviewer.

  1. In the section 2.2, please clearly mention what are the inclusion and exclusion criteria? Please also justify why this study is conducted in Asir? Any particular reason?

Response to reviewers: The inclusion criteria: Qualified pharmacists working in different pharmacy sectors in Asir region, Saudi Arabia. Exclusion criteria: Pharmacy technicians, unemployed pharmacists, pharmacists working outside Asir region. The study was conducted in the Asir region only due to accessibility. 

Changes in the manuscript: Methodology: The inclusion criteria: Qualified pharmacists working in different pharmacy sectors in Asir region, Saudi Arabia. Exclusion criteria: Pharmacy technicians, unemployed pharmacists, pharmacists working outside Asir region.

  1. In the section 2.3, please write what was the sampling method? Was it random probability sampling or non-probability sampling? Which type?

Response to reviewers: Non-probability sampling. A convenience sampling strategy was used to select and recruit the study participants. 

  1. In the section 2.5, please write “data collection tool” instead of ‘data collection form”.

Response to reviewers: Changes done as Data collection tool

  1. In the section 2.5, please write how did you develop the questionnaire? Was it already developed? Or you generate it suing other similar questionnaire?

Response to reviewers: The questionnaire was adapted from previous studies (18,19).

  1. Please write about study procedure. How did you reach the study participants? How long it took time the questionnaire to be completed? How about the period (time interval) to complete the questionnaire? How about non-respondent rate?

Response to reviewers: The sampling strategy used in this research was convenience sampling. Participants were invited to participate in the study through multiple channels i.e., official institutional emails and social media platforms. Monthly reminders were sent to the potential participants with an attempt to maximize the response rate. 

  1. Please present the results of Figure 1 in a table.

Response to reviewers: Results of Figure 1 has been presented in table 5

Table 5. CME activities by pharmacy sector.

CME activities by pharmacy sector

Relevant to practice

Based on personal needs

CME rewarded

Achieving hours

Community Pharmacy

77.78

72.22

72.22

83.33

Hospital Pharmacy

78.08

79.45

72.6

78.08

Academia

82.98

72.34

74.47

59.57

Regulatory

66.67

77.78

77.78

77.78

Industry

75

62.5

62.5

75

  1. Please write about lack of generalizability of finding to all Saudi Arabia, as one of the limitations of the study due to nature of study design

Response to reviewers: Thank you for your comment. We added this point as a limitation in the discussion section. 

Changes in the manuscript: Additionally, due to the nature of the study design, the generalizability of the findings is limited to those with similar context. 

Once again, thank you to all the reviewers for the valuable comments and suggestions. We have revised our manuscript strictly and thoroughly as per your comments for more clarity to the reader.

Reviewer 3 Report

The authors share the results of a simple questionnaire survey of pharmacists' participation in CME. I could not find any surprises in terms of the findings and their interperetation. What makes the draft interesting is that it reports on experiences in Saudi Arabia, and little information was previously available on the details of public health in this country. It is reassuring that in Saudi Arabia the internationally accepted continuing education requirements apply in this profession as well, and that catching up with new trends is also on the agenda (CPD model), and if I understand correctly from what has been described, this is similar in other Gulf countries. The writing meets the basic expectations, although a comparative review with other Arab states could have been more exciting. Although I recommend its adoption, editors should decide whther publication should receive "green light" mostly because of the scene of the survey. 

Author Response

Thank you for your valuable comments. We have revised our article thoroughly as per your comments. The individual response to reviewer comments are given below.

Manuscript ID: Healthcare-2486693

Reviewer 3 comments

The authors share the results of a simple questionnaire survey of pharmacists' participation in CME. I could not find any surprises in terms of the findings and their interpretation. What makes the draft interesting is that it reports on experiences in Saudi Arabia, and little information was previously available on the details of public health in this country. It is reassuring that in Saudi Arabia the internationally accepted continuing education requirements apply in this profession as well, and that catching up with new trends is also on the agenda (CPD model), and if I understand correctly from what has been described, this is similar in other Gulf countries. The writing meets the basic expectations, although a comparative review with other Arab states could have been more exciting. Although I recommend its adoption, editors should decide whether publication should receive "green light" mostly because of the scene of the survey. 

Once again, thank you to all the reviewers for the valuable comments and suggestions. We have revised our manuscript strictly and thoroughly as per your comments for more clarity to the reader.

Round 2

Reviewer 1 Report

The authors revised the manuscript regarding the reviewer's suggestions. It can be accepted.

Reviewer 2 Report

Thank you for considering all my comments. 

That is acceptable.